# Research on an Enhanced Multimodal Network for Specific Emitter Identification

Heli Peng [ID], Kai Xie * and Wenxu Zou [ID]

Department of Electronics and Communication Engineering, Sun Yat-Sen University, Shenzhen 518107, China; penghli@mail2.sysu.edu.cn (H.P.)
* Correspondence: xiek8@mail.sysu.edu.cn

**Abstract:** Specific emitter identification (SEI) refers to the task of distinguishing similar emitters, especially those of the same type and transmission parameters, which is one of the most critical tasks of electronic warfare. However, SEI is still a challenging task when a feature has low physical representation. Feature representation largely determines the recognition results. Therefore, this article expects to move toward robust feature representation for SEI. Efficient multimodal strategies have great potential for applications using multimodal data and can further improve the performance of SEI. In this research, we introduce a multimodal emitter identification method that explores the application of multimodal data, time-series radar signals, and feature vector data to an enhanced transformer, which employs a conformer block to embed the raw data and integrates an efficient multimodal feature representation module. Moreover, we employ self-knowledge distillation to mitigate overconfident predictions and reduce intra-class variations. Our study reveals that multimodal data provide sufficient information for specific emitter identification. Simultaneously, we propose the CV-CutMixOut method to augment the time-domain signal. Extensive experiments on real radar datasets indicate that the proposed method achieves more accurate identification results and higher feature discriminability.

**Keywords:** multimodal model; specific emitter identification; transformer; conformer; data augmentation; class-wise self-knowledge distillation



## 1. Introduction

Radar specific emitter identification (SEI) is an important electronic intelligence (ELINT) activity that aims to recognize individual emitters using features related to the so-called unintentional modulation on pulse (UMOP) [1]. These subtle features are called the radio frequency (RF) fingerprint, are mainly caused by hardware defects in the manufacturing process, and play an important role in fields such as electronic reconnaissance and physical-layer security [2]. Consequently, fingerprint feature representation is the key to SEI.

The traditional SEI scheme is mainly based on manual feature extraction, which classifies the signal by extracting the RF fingerprint features of the signal such as the pulse width (PW), pulse repetition interval (PRI), radio frequency (RF), antenna scan type/period (AST/ASP), and the intentional modulation on pulse (IMOP) [1]. However, in modern electronic warfare (EW), identifying emitters based on traditional characteristics becomes an increasingly challenging task. Due to the rapid development of deep learning (DL), neural networks have achieved better performance at identification [3–6], but they are mostly considered in single-modal scenarios. Multimodal data, such as feature vectors, time-frequency diagrams, bispectral diagrams, etc., contain sufficient information. By effectively utilizing multimodal data, SEI can achieve better results and can be applied to few-shot learning [7–9].

Most prior efforts in this domain have focused on extracting the features of time-domain signals or converting signals into images for further feature extraction. Aubry

et al. [10] used the cumulants of the signal emitted by the radar system as the input to a k-nearest neighbor (KNN) classifier. D'Agostino et al. [11] discussed basic subtle features including the rise time, fall time, and overshoot of a square wave. Ding et al. [12] proposed a deep-learning-based SEI approach that uses the features of the received steady-state signals. Particularly, they calculated the bispectrum of the received signals as a feature. Zhu et al. [13] proposed a compressed sensing mask feature in the ambiguity domain, which can significantly improve the recognition rate of civil flight radar emitters. While this research direction has remained popular in recent decades, its dependence on expert experience and large, complex models pose challenges for specific emitter identification in real scenarios.

Recently, there has been a growing interest in exploring deep multimodal learning for both time-domain data and feature data with the aim of constructing a common representation space. Multimodal learning models, such as [3,5], have enabled derivation of data–feature representations that improve the SEI task. Concurrently, since multimodal data depict an object from different viewpoints, usually complementary or supplementary in content, they are more informative than unimodal data [14]. So multimodal learning is also widely used in other fields. Bharadwaj et al. [15] proposed MuSeLI, a multimodal spoken language identification method, which delves into the use of various metadata sources to enhance language identification. Tu et al. [16] introduced a dilated convolutional transformer for modeling and estimating human engagement. Urbanelli et al. [17] proposed a novel multimodal supervised machine learning approach to disambiguate hotspot detection in order to distinguish between wildfires and other events. Multimodal learning has great potential not only in SEI but also in medical image processing, speech recognition, video annotation, and other fields.

Based on the above analysis, we introduce a multimodal learning approach designed to enhance specific emitter identification. Inspired by [15], the proposed method utilizes a transformer network enhanced by the conformer layers to generate data embedding. And we use the weighted layer representation to implement multimodal data representation. In addition, class-wise self-knowledge distillation is used in training to mitigate overconfident predictions and reduce intra-class variations. Aiming to enhance the performance and robustness of deep learning models, we involve the use of new data augmentation techniques: "CV-CutMixOut" from "CutMixOut" [18]. In summary, the main contributions of this paper are as follows.

- We propose a multimodal SEI model based on an improved transformer network that facilitates the incorporation of diverse data called MuSEI, **Mu**ltimodal **S**pecific **E**mitter **I**dentification. By utilizing conformer layers [19], which are a combination of self-attention and convolution to achieve global and local interactions, MuSEI can obtain better recognition accuracy. A multimodal representation module employing weighted layer representation has shown remarkable efficacy at merging multiple modalities. Deep neural networks with millions of parameters might experience poor generalization due to overfitting. In addressing this issue, this study employs a new regularization technique, coined class-wise self-knowledge distillation (CSKD), which, compared to cross-entropy loss, mitigates overfitting to a certain extent.
- We introduce a new data augmentation method: "CV-CutMixOut". It integrates k-fold cross-validation and CutMixOut to fortify data robustness while minimizing information loss.
- Extensive experiments on a real dataset are presented to evaluate the performance of MuSEI. Our proposed method achieves state-of-the-art results, outperforming the previous public benchmarks.

The paper is arranged as follows: Section 2 discusses the related work on specific emitter identification and categorizes it into two parts: traditional methodologies and deep-learning-based approaches. Section 3 introduces an efficient multimodal model for SEI and an enhanced data augmentation scheme. Section 4 shows and discusses the results of comparative experiments on real datasets. Section 5 gives the conclusions.

## 2. Related Works

Specific emitter identification has consistently remained a pivotal technology within electronic reconnaissance, which itself holds paramount importance in the domain of electronic warfare. Through the extraction of subtle features, often arising from variances in hardware circuitry such as modulators, power amplifiers, and transmitters, within radio frequency signals, radar emitters are identified. Previous research relied on manual feature extraction [20–22]: encompassing attributes like pulse width, pulse repetition interval (PRI), leading-edge slope, and rise time. However, this approach heavily relies on expert experience. As the electromagnetic signal density continues to rise and the electromagnetic environment becomes more intricate, depending solely on expert knowledge for feature extraction diminishes adaptability. This presents a significant challenge for individual identification of emitters.

With the substantial advancements in deep learning, its pervasive application spans across domains like computer vision, natural language processing, and speech recognition. Deep learning adeptly achieves diverse levels of feature representation and knowledge abstraction. In SEI, leveraging unintentional modulation on pulse (UMOP) features extracted via existing analysis mechanisms and inputting them into a deep network allows for progressive enhancement of feature representation through hierarchical learning. This process aims to elevate recognition performance, potentially attenuating the association between individual feature generation mechanisms and extraction methods. Consequently, the individual identification of radar emitters based on deep learning methods has emerged as a prominent research focal point in recent years. O'Shea demonstrated that semi-supervised learning techniques can be used to scale learning beyond supervised datasets to allow for discerning and recalling new radio signals by using sparse signal representations based on both unsupervised and supervised methods for nonlinear feature learning and clustering methods. Roy et al. [23] implemented a generative model that learns the sample space of the I/Q values of known transmitters and uses the learned representation to generate signals that imitate the transmissions of these transmitters. Apfeld et al. [24] investigated six approaches using several configurations to recognize unknown emitters based on a hierarchical emission model that understands emissions as a language with an inherent hierarchical structure. Sankhe et al. [25] presented a novel system based on convolutional neural networks (CNNs) to identify a unique radio from a large pool of devices by deep learning the fine-grained hardware impairments imposed by radio circuitry on physical-layer I/Q samples. Tan et al. [26] introduced semi-supervised learning into SEI and proposed a self-classification generative adversarial network (GAN) using bispectrum-based feature extraction. The aforementioned approaches can learn the inherent features of different emitters: underscoring the ascendancy of deep learning in SEI. Yet these methodologies mainly utilize single-modal data.

Recently, there have been an increasing number of studies exploring joint modeling techniques for multimodal data with the aim of constructing a shared space for multimodal representation. Satija et al. developed emitter identification based on variational mode decomposition and spectral features (VMD-SF). They evaluated the performance of the proposed methods using the probability of correct classification both in single-hop and in relay scenarios by varying the number of emitters. Guo et al. [14] provided a comprehensive survey of deep multimodal representation learning. They categorize deep multimodal representation learning methods into three frameworks: joint representation, coordinated representation, and encoder–decoder. Urbanelli et al. [17] proposed a novel multimodal supervised machine learning approach to disambiguate hotspot detection to distinguish between wildfires and other events. Tu et al. [16] introduced a dilated convolutional transformer for modeling and estimating human engagement by employing the modalities of the three attributes as the signal. Through fusing multiple modalities, these methodologies substantiate the extensive utilization and promising development prospects of multimodal networks across various domains [17,27,28]. Multimodal representation is gradually being

widely utilized in fields such as natural language processing, speech recognition, medical impact analysis, and intelligent transportation [15–18,27].

Overall, current SEI has moved from transient features to steady-state features; from linear, hand-crafted transformation to nonlinear, data-driven representation; and from single-feature to multimodal transformation fusion [3]. In [3], the SEI problem is considered in single-modal, dual-modal, and multimodal scenarios, respectively. Considering the similarity between radar time-domain signals and signals such as speech and EEG, our endeavor involves employing a multimodal model for specific emitter identification. This approach aims not only to maximize the utility of collected radar time-domain signals but also to proficiently fuse other pertinent information.

## 3. Methods

In this section, we introduce our method in two parts. The first part is to introduce the "CV-CutMixOut" data augmentation method. Then, we introduce a multimodal model MuSEI to learn multimodal representation of radar time-domain data and feature inputs.

### 3.1. Data Augmentation

To enhance the performance and robustness of deep learning models for the SEI task, we propose a data augmentation method for radar time-domain data, which is improved from [18]. Image augmentation techniques, specifically cutout and cutmix, are commonly used for computer vision tasks and randomly mask a portion of an image. Fawakherji et al. [18] apply these techniques to the text portion of the query input to create multiple representations of the text. Considering the similarity between speech input and time-domain signal data, we adopt a "CV-CutMixOut (cross-validated CutMixOut)" method: amalgamating k-fold cross-validation and "CutMixOut" techniques.

"CutMixOut" merges the two augmentation strategies "cutout" and "cutmix" into one strategy. To achieve cutmix, a binary mask $\mathbf{M} = [m_1, m_2, \ldots, m_n]$ of length $n$ is used to select a contiguous subsequence in raw signal $\mathbf{S}$, for which the subsequences are substituted with their corresponding parts from a shuffled duplicate of $\mathbf{S}$. Note that the raw signal $\mathbf{S}$ here and afterward refers to the envelope signal obtained by the Hilbert transform of the collected radar signal. The results can be calculated as:

$$\mathbf{S}_{CutMix} = \mathbf{M} * \mathbf{S} + (1 - \mathbf{M}) * \mathbf{S}' \tag{1}$$

where $\mathbf{S}$ is raw data of length $n$, and $S'$ is a shuffled version of $\mathbf{S}$.

Likewise, adhering to the principles of cutout, which involves the random removal of segments from input data $\mathbf{S}$, enables the generation of a more robust dataset for training. Similarly, the aforementioned binary mask $\mathbf{M}$ remains applicable, wherein $m_i = 0$ denotes the removal of the input and $m_i = 1$ otherwise. The new data $\mathbf{S}_{CutOut}$ are calculated as:

$$\mathbf{S}_{CutOut} = \mathbf{M} * \mathbf{S} \tag{2}$$

Then, the CutMixOut is obtained by randomly choosing between $\mathbf{S}_{CutMix}$ and $\mathbf{S}_{CutOut}$:

$$\mathbf{S}_{CutMixOut} = \begin{cases} \mathbf{S}_{CutMix} & \text{with probability } p_{CutMix} \\ \mathbf{S}_{CutOut} & \text{with probability } p_{CutOut} \end{cases} \tag{3}$$

Here, $p_{CutMix}$ and $p_{CutOut}$ are assigned a certain probability for each operation. However, random cropping or replacement that is too extensive will lead to severe loss of subtle features, thereby affecting recognition accuracy. Therefore, we use k-fold cross-validation to find the appropriate size of CutMixOut in contrast to random processing of the raw data: a methodology referred to as "CV-CutMixOut". The implementation steps are as follows:

1. Split the data into $k$ parts.
2. Try different CutMixOut sizes: size $\mathbf{S}_{com} = [com_1, com_2, \ldots, com_l]$, where $l$ is the number of selectable sizes and $com_i$ is the number of $m_i = 0$.

3. For each size, we utilize one subset of the data as the test set and the rest as the training set to capture the model's performance.
4. Ultimately, we compare the model's performance across all sizes to determine the most suitable size.

### 3.2. MuSEI: Improved Multimodal Framework for SEI

A comprehensive overview of the proposed multimodal emitter identification system is shown in Figure 1. MuSEI is based on [15], which processes speech and text multimodal language recognition systems. The MuSEI framework is composed of three modules, i.e., a raw data embedding module, a multimodal representation module, and a classification module composed of transformer layers and a classifier. Sections 3.2.1 and 3.2.2 introduce the main two modules, respectively. Section 3.2.3 introduces the improved loss function with self-knowledge distillation. The raw data depicted in Figure 1 mean the time-series envelope signal, while the feature vector is constructed from features extracted from the envelope signal, $F_i, i = 1, 2, 3 \ldots, k$ denotes the feature parameter, and k is the number of feature parameters. The transformer layers are composed of a stack of $N$ identical layers.

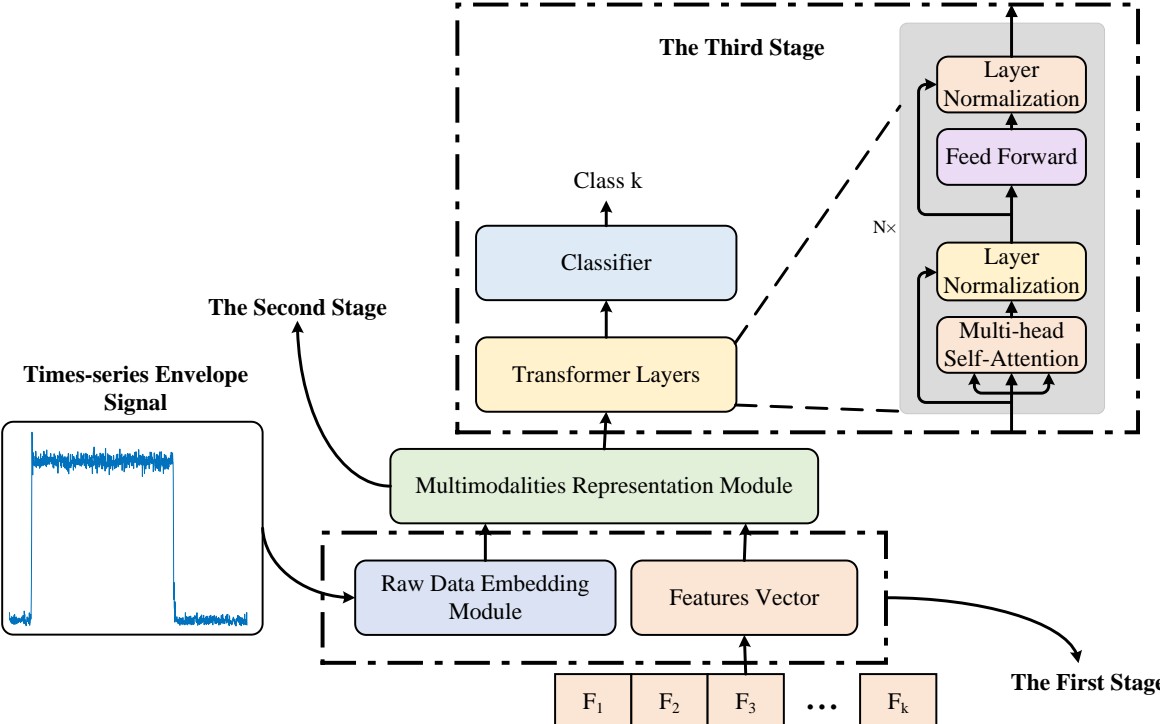

**Figure 1.** The overview of the proposed method.

The whole process of this method mainly consists of three parts: firstly, employing the proposed data enhancement technique "CV-CutMixOut" to fortify both the robustness of the data and the learning efficacy of the network; secondly, utilizing the raw data embedding module to conduct latent representation; lastly, via a multimodal representation module, raw data representation and feature representation are amalgamated to construct a unified representation, facilitating SEI by employing transformer layers consisting of multi-headed self-attention (MSA) [29], layer normalization (LN) [30], and MLP blocks [31].

### 3.2.1. Raw Data Embedding Module

Transformers excel at capturing long-range global contexts but struggle with extracting precise local feature patterns. Conversely, convolution neural networks (CNNs) leverage local data and serve as the de facto computational unit in computer vision tasks. They harness position-based kernels over specific windows: preserving translation equivariance

and adeptly capturing features such as edges and shapes [19]. However, relying solely on local connectivity necessitates numerous additional layers or parameters to effectively capture global information. Considering this, we propose the use of a conformer module to embed the raw data.

Our raw data embedding module, which consists of a stack of conformer blocks, initially processes the input to produce a latent signal representation. The conformer block is illustrated in Figure 2. A conformer block is composed of four modules stacked together: a feed-forward module, a self-attention module, a convolution module, and a second feed-forward module at the end [19].

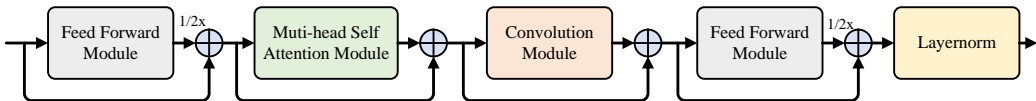

**Figure 2.** The conformer block.

The feed forward-module, which is also adopted by the transformer [29], is composed of two linear transformations and a nonlinear activation in between. A residual connection is added over the feed-forward layers, followed by layer normalization. The convolution module contains a pointwise convolution with an expansion factor of two projecting the number of channels with a gated linear unit (GLU) activation layer, followed by a 1-D depthwise convolution. Batchnorm is deployed after the 1-D depthwise convolution to aid training deep models.

Mathematically, the conformer block is as follows:

$$
\begin{aligned}
\widetilde{x}_i &= x_i + \frac{1}{2}FFN(x_i) \\
x_i' &= \widetilde{x}_i + MHSA(\widetilde{x}_i) \\
x_i'' &= x_i' + Conv(x_i') \\
y_i &= LayerNorm(x_i'' + \frac{1}{2}FFN(x_i''))
\end{aligned}
\tag{4}
$$

where, $x_i$ is input to a conformer block $i$, and $y_i$ is the output of the block.

The raw data **S** undergo processing through the raw data embedding module $F_{RDE}$ to produce latent signal representation $L$. It is computed as:

$$
\mathbf{L} = F_{RDE}(\mathbf{S})
\tag{5}
$$

where RDE means the raw data embedding module.

As shown in Figure 1, the first stage of this method is raw data embedding and estimation of feature vectors. Next, we introduce the establishment of feature vectors. Feature vectors are extracted from the original signals. Guo et al. [32] proposed to use the radio frequency (RF) features, including the duration, maximum derivative, skewness, kurtosis, mean, variance, fractal dimension, Shannon entropy, and polynomial coefficients of the normalized energy trajectory of a transient signal, as well as the area under the trajectory curve extracted from time-domain transient signals for radar model identification. The results show that the proposed AF fingerprint can be applied directly to radar model identification.

In specific emitter identification, a raw signal of length $n$ is $\mathbf{S} = [s_1, s_2, \ldots, s_n]$, and the feature vector composed of signal rising edge time $T_{rise}$, mean $\mu$, variance $\sigma$, fractal dimension $frac$, Shannon entropy $en$, kurtosis $kappa$, and skewness $sk$ extracted from the original signal **S** is $\Phi = [T_{tise}, \mu, \sigma, frac, en, kappa, sk]$. The calculation formulas for $\mu$, $\sigma$, $kappa$, and $sk$ are shown in Table 1. The calculation formulas of the fractal dimension and Shannon entropy are complicated; they are not listed here. The fractal dimension and Shannon entropy of a one-dimensional signal are important metrics to assess distinct aspects of the signal's characteristics. The fractal dimension measures the complexity and

regularity of a signal or dataset. Shannon entropy gauges the uncertainty or information content of a signal.

**Table 1.** Feature parameter calculation formulas.

| Feature Parameter | Calculation Formula |
|---|---|
| mean $\mu$ | $\mu = \dfrac{\int_{-\infty}^{\infty} t|u(t)|^2 dt}{\int_{-\infty}^{\infty} |u(t)|^2 dt}$ |
| variance $\sigma$ | $\sigma = \left( \dfrac{\int_{-\infty}^{\infty} (t-\mu)^2 |u(t)|^2 dt}{\int_{-\infty}^{\infty} |u(t)|^2 dt} \right)^{\frac{1}{2}}$ |
| kurtosis *kappa* | $\kappa = \dfrac{\int_{-\infty}^{\infty} (t-\mu)^4 / \sigma^4 |u(t)|^2 dt}{\int_{-\infty}^{\infty} |u(t)|^2 dt}$ |
| skewness *sk* | $sk = \dfrac{\int_{-\infty}^{\infty} (t-\mu)^3 / \sigma^3 |u(t)|^2 dt}{\int_{-\infty}^{\infty} |u(t)|^2 dt}$ |

### 3.2.2. Multimodal Representation Module

Simultaneously, Hsu et al. [33] demonstrated that the final layer representation acquired after multiple layers might not be optimal for all tasks. Consequently, we introduce a learnable parameter $\beta_k$ in training, which corresponds to the $k$th conformer layer. It is calculated by:

$$\mathbf{L} = \sum_k \beta_k \mathbf{L}_k \tag{6}$$

where $\mathbf{L}_k$ is the representation from the $k_{th}$ conformer layer.

To facilitate the merging of raw signal data and feature information, we employ a multimodal representation module, as illustrated in Figure 3. We use the weighted combination of embeddings from the conformer blocks in Equation (6). Then, the concatenated time-series signal and feature vectors are passed to the transformer layers. That is, firstly, the raw data representation $\mathbf{L}$ and the feature vector $\Phi$ are concatenated to be $\mathbf{R} = [\mathbf{L}; \Phi]$. Then, the concatenation $\mathbf{R}$ is input to the transformer layers [29] to capture the significance of each modality effectively.

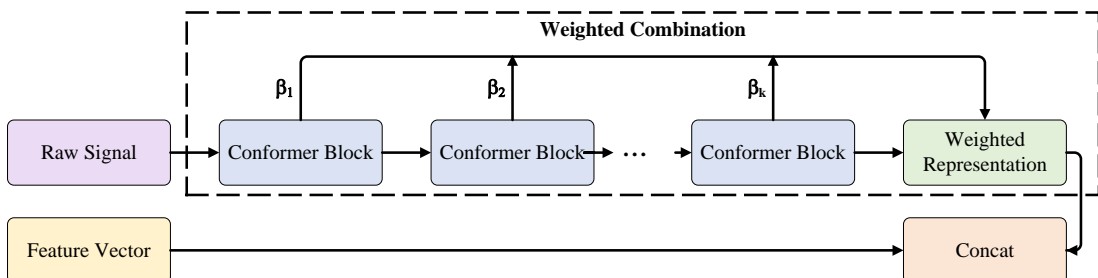

**Figure 3.** The multimodal representation module.

### 3.2.3. The Improved Loss

To mitigate overfitting and minimize the intra-class variations, the module is trained using the combination of the cross-entropy loss and the class-wise self-knowledge distillation (CSKD) loss [34]. The cross-entropy loss is to enhance the discrimination between different classes. The CSKD loss distills the predictive distribution of network between different samples of the same class, as illustrated in Figure 4.

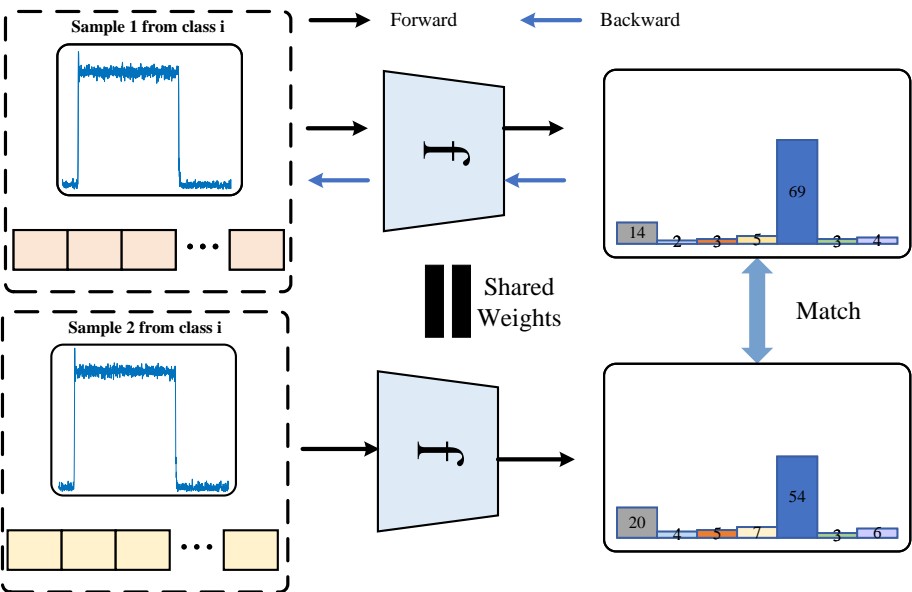

**Figure 4.** The CSKD regularization scheme.

In Figure 4, sample 1 and sample 2 come from the same class. By matching the predictive distribution of the network between different samples with the same label, CSKD forces the network to produce similar predictions, thereby achieving the goals of preventing overfitting and reducing the intra-class variations. The network samples a batch from the training dataset, serving as sample 1, and extracts another batch from the training dataset randomly with the same label, forming sample 2. Then, we use Equation (9) to update the gradient until the parameters converge.

In the proposed method, given the input $\mathbf{x}$ and class $y$, $S \in \mathbf{x}$, $y \in 1, 2, \ldots, k$, the posterior predictive distribution is:

$$P(y|\mathbf{x}; \theta, T) = \frac{exp(f_y(\mathbf{x}; \theta)/T)}{\sum_{i=1}^{k} exp(f_i(\mathbf{x}; \theta)/T)} \tag{7}$$

where $f_i$ denotes the logit of networks for class $i$, which is parameterized by $\theta$, and $T > 0$ is the temperature scaling parameter [34].

To enforce consistent predictive distributions in the same class, we obtain randomly sample $\mathbf{x}'$ with the same label $y$ as the input $\mathbf{x}$. The calculation is as follows:

$$\mathcal{L}_{cls}(\mathbf{x}, \mathbf{x}'; \theta, T) = KL(P(y|x'; \widetilde{\theta}, T)||P(y|\mathbf{x}; \theta, T)) \tag{8}$$

where KL denotes the Kullback–Leibler (KL) divergence, and $\widetilde{\theta}$ is a fixed copy of the parameter $\theta$. To sum up, the total loss function is:

$$\mathcal{L}_{CS-KD}(\mathbf{x}, \mathbf{x}', y; \theta, T) = \mathcal{L}_{CE}(\mathbf{x}, y; \theta) + \lambda_{cls} \cdot T^2 \cdot \mathcal{L}_{cls}(\mathbf{x}, \mathbf{x}'; \theta, T) \tag{9}$$

where $\mathcal{L}_{CE}$ is the standard cross-entropy loss, and $\lambda_{cls} > 0$ is a loss weight for the class-wise regularization.

## 4. Experiments

### 4.1. Datasets and Parameter Settings

Our experimental system is Windows 11. The experimental setup involves PyTorch 1.2, a widely used deep learning framework, implemented using Python 3.9.

To evaluate our method, we use a real radar dataset consisting of data from six emitters collected in the same environment for recognition tasks. The carrier frequency of the signal is 9300 MHz, and the sampling frequency is 500 MHz. The dataset contains six types of

single-frequency signal samples. It contains 3000 radar signals and 3000 feature vectors extracted from the radar signals. The dataset is split into a training set and a test set in a ratio of 7:3. Each radar signal is segmented into a length of $1 \times 1024$, and the feature vector comprises a length of $1 \times 7$. We randomly select a sample with different labels from the dataset to display. The original data are illustrated in Figure 5. The envelope data are illustrated in Figure 6.

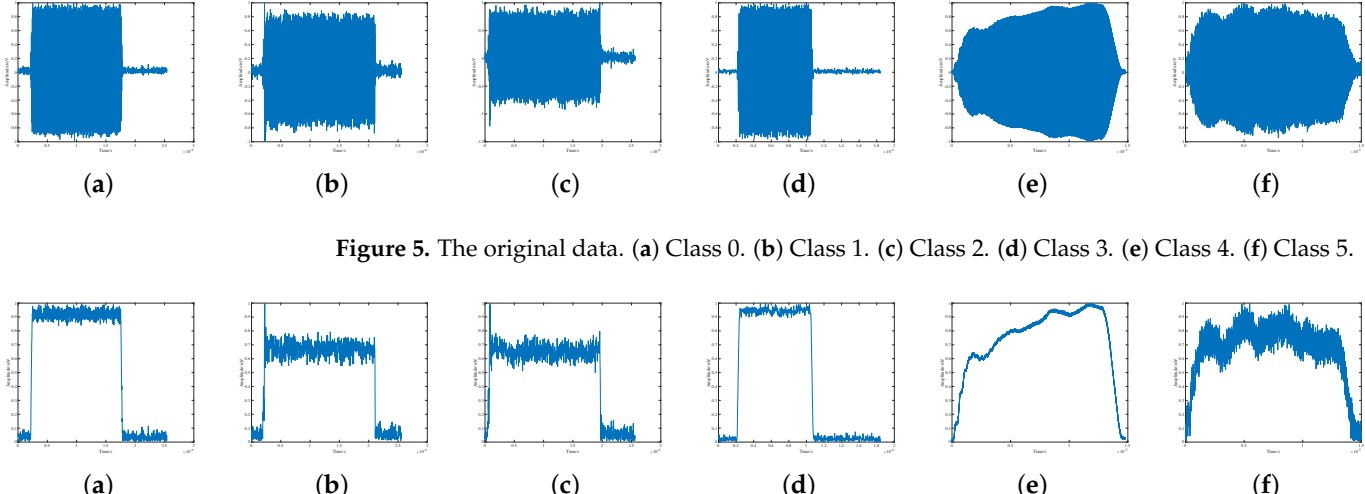

(**a**)     (**b**)     (**c**)     (**d**)     (**e**)     (**f**)

**Figure 5.** The original data. (**a**) Class 0. (**b**) Class 1. (**c**) Class 2. (**d**) Class 3. (**e**) Class 4. (**f**) Class 5.

(**a**)     (**b**)     (**c**)     (**d**)     (**e**)     (**f**)

**Figure 6.** The envelope data. (**a**) Class 0. (**b**) Class 1. (**c**) Class 2. (**d**) Class 3. (**e**) Class 4. (**f**) Class 5.

Figures 5 and 6 show the original pulse data and the corresponding envelope data. Some classes, such as class 0 and class 2, are difficult to separate since their envelopes are similar. Additionally, while class 4 and class 5 differ from others, distinguishing between them remains difficult. The fractal dimension is one of the most used and significant features in SEI [32,35–37]. Here, we provide the fractal dimension distribution. Figure 7 illustrates the distribution of fractal dimension features extracted from envelope data. A box plot, also known as a box-and-whisker plot, is a graphical representation that displays the summary of a set of data values. It provides a visual summary of the distribution and key statistical measures of a dataset, including the median, quartiles, and potential outliers. On each box, the center mark represents the median, and the bottom and top edges of the box represent the 25th and 75th percentiles, respectively. The line will extend to the farthest data point that is not an outlier, and the outlier will be drawn separately using the red '+' symbol.

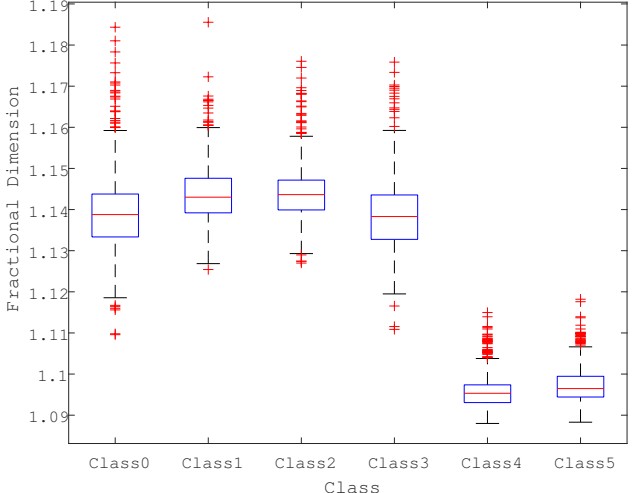

**Figure 7.** The boxplot representation of fractal dimension for different emitters.

Emitters of different radars cannot be effectively identified using a single feature parameter. From Figure 7, it is evident that there are overlapping regions in the feature parameter value distribution, and they are relatively dispersed. Moreover, there are feature parameters from different samples in the same class distributed in different ranges, which also leads to difficulty in identification.

Firstly, we present the experimental results produced by using feature vector datasets with SVM algorithms. The SVM algorithm is a supervised machine learning algorithm for binary classification [38]. It is a representation of the training data as points in space separated into classes by a clear gap that is as wide as possible. New data are then mapped into that same space and are predicted to belong to a class based on which side of the gap they fall on [32]. SVM is implemented in MATLAB (The MathWorks Inc., *MATLAB.* v.9.12.0 R2022a, Natick, MA, USA)) toolboxes. The confusion matrix in Figure 8 shows the classification accuracy for each emitter in the six-emitter mix.

A confusion matrix is a way of describing the breakdown of classification accuracy for a testing dataset. It makes the performance evaluation easier as one can see if a classifier is confusing emitters [32]. In the confusion matrix, the rows (except the bottom one) correspond to the classified emitters and the columns (except the far right one) correspond to the actual emitters. The column on the far right of a confusion matrix shows the percentages of all feature vectors predicted to belong to each classified emitter that are correctly or incorrectly classified. The row at the bottom shows the percentages of all feature vectors belonging to each actual emitter that are correctly or incorrectly classified. The cell on the bottom right of a confusion matrix shows the overall classification accuracy [32].

**Confusion Matrix of SVM with feature vectors**

|   | 0 | 1 | 2 | 3 | 4 | 5 | |
|---|---|---|---|---|---|---|---|
| **0** | **145**<br>16.1% | 4<br>0.4% | 12<br>1.3% | 11<br>1.2% | 1<br>0.1% | 0<br>0.0% | 83.8%<br>16.2% |
| **1** | 15<br>1.7% | **243**<br>27.0% | 17<br>1.9% | 14<br>1.6% | 19<br>2.1% | 13<br>1.4% | 75.7%<br>24.3% |
| **2** | 2<br>0.2% | 12<br>1.3% | **56**<br>6.2% | 5<br>0.6% | 9<br>1.0% | 2<br>0.2% | 65.1%<br>34.9% |
| **3** | 18<br>2.0% | 12<br>1.3% | 12<br>1.3% | **54**<br>6.0% | 0<br>0.0% | 0<br>0.0% | 56.2%<br>43.8% |
| **4** | 1<br>0.1% | 9<br>1.0% | 3<br>0.3% | 3<br>0.3% | **70**<br>7.8% | 9<br>1.0% | 73.7%<br>26.3% |
| **5** | 0<br>0.0% | 8<br>0.9% | 1<br>0.1% | 3<br>0.3% | 10<br>1.1% | **107**<br>11.9% | 82.9%<br>17.1% |
| | 80.1%<br>19.9% | 84.4%<br>15.6% | 55.4%<br>44.6% | 60.0%<br>40.0% | 64.2%<br>35.8% | 81.7%<br>18.3% | **75.0%**<br>**25.0%** |

Classified Emitters (vertical axis) / Actual Emitters (horizontal axis: 0, 1, 2, 3, 4, 5)

**Figure 8.** Confusion matrix for the SVM algorithm with feature vector for six-emitter mix. The green cells show classification accuracy and number of correctly classified feature vectors for corresponding actual emitter. Meanwhile, the red cells indicate the number of misclassified vectors from each actual emitter for each classified emitter as well as its percentage of the total number of tested feature vectors.

Emitter 2 has the lowest classification accuracy of 55.4%, while emitter 1 has the highest classification accuracy of 84.4%. It can be seen that with the real feature vector

dataset, the method with the SVM algorithm only achieves on average 75% classification accuracy. The results demonstrate that it is hard to classify a real radar dataset and that the feature vector dataset alone is inadequate for classifying and identifying real radar emitters.

In addition, we simulated a dataset, called dataset II, with differences in pulse width, rising edge slope, rising edge time, signal type, etc. Dataset II contains 5000 radar signals in ten classes. The signal styles include linear frequency modulation signals, single-frequency signals, etc. We add additive Gaussian white noise to the samples to further simulate the influence of various additive noises on signals.

Then, the multimodal dataset is used to validate if the proposed method is suitable for SEI. The parameter settings during the training are shown in Table 2. The learning rate is a hyperparameter that controls the magnitude of parameter updates during the training process. Batch size refers to the number of samples that are processed by the model in a single iteration during training. An epoch signifies one complete pass of the entire dataset through the neural network during the training phase. Weight decay, often used as a form of regularization, introduces a penalty term to the loss function based on the magnitude of the model weights. An optimizer is an algorithm that adjusts the model's weights during training to minimize the loss function.

**Table 2.** Training parameter configuration.

| Signal Parameter | Parameter Value |
|---|---|
| Learning rate | 0.0001 |
| Batch Size | 64 |
| Epoch | 300 |
| Weight Decay | 0.0001 |
| Optimizer | AdamW |

*4.2. Performance Comparison*

In experiments, the effectiveness of data augmentation "CV-CutMixOut" and multi-modal MuSEI are tested. Experiment 1 evaluates the effectiveness of data augmentation on recognition results. Experiment 2 assesses the performance of different models. Experiment 3 conducts comparisons using different loss functions.

4.2.1. Experiment 1: "CV-CutMixOut" effect on Recognition Performance

Based on the aforementioned analysis, our initial experiment aims to evaluate the effectiveness of the data augmentation method for the SEI task. In MuSEI, the experiment incorporates multi-modal inputs, whereas other one-dimensional networks exclusively utilize time-domain radar signals as their input. The results are shown in Table 3. We use the following models to test our method: RNN, LSTM, 1DCNN, transformer, and MuSEI. Each of these models is evaluated both with and without data augmentation. The sizes for CutMixOut we compared are $\mathbf{S}_{com} = [50, 100, 150, 200, 250]$. Simultaneously, all networks in Table 3 employ CSKD as their loss function. The results show that all models achieve higher top-k accuracies with data augmentation than without. The largest improvement is seen in the RNN model, which achieves a top-1 accuracy of 84.56% with data augmentation compared to 73.21% without. The RNN model and LSTM show a significant improvement in the top-3 accuracy with data augmentation. MuSEI shows less improvement, likely due to its performance reaching saturation or model capacity limits. With data augmentation techniques, both the transformer network and MuSEI achieved notable advancements in recognition accuracy. Moreover, compared to scenarios without leveraged data augmentation, the network's recognition accuracy exhibited substantial improvement.

**Table 3.** Performance of different models with (✓) and without (×) data augmentation.

| Model | CV-CutMixOut | Top 1 (%) | Top 3 (%) |
|---|---|---|---|
| RNN | × | 73.21 | 87.41 |
| | ✓ | 84.56 | 92.78 |
| LSTM | × | 80.69 | 90.70 |
| | ✓ | 89.78 | 95.77 |
| 1DCNN | × | 81.45 | 95.43 |
| | ✓ | 90.39 | 96.20 |
| Transformer | × | 82.21 | 95.31 |
| | ✓ | 91.52 | 96.54 |
| MuSEI | × | 92.01 | 98.87 |
| | ✓ | **96.67** | 99.02 |

4.2.2. Experiment 2: Identification Based on MuSEI on Real Dataset

In this experiment, we conduct comparisons based on the recognition accuracy of different models, including single-modal networks and multimodal networks. Moreover, we evaluate the effectiveness of each module in MuSEI.

Firstly, we evaluate the performance of different models for the SEI task. Figure 9 shows the loss curves for the training iterations. All models employ data augmentation techniques in conjunction with the CSKD loss function. The enhanced multimodal model achieves the highest performance: the fastest convergence of the loss curves. Furthermore, as depicted in Table 3, the results show that it achieves the highest recognition accuracy. The MuSEI loss converges to about 0.5, which indicates a strong fit between the model and the data, showcasing the model's adeptness at capturing intricate data characteristics and patterns. The pronounced fluctuation in the convergence of the loss curve in the RNN network may stem from its heightened sensitivity to imbalanced samples. The loss convergence values of the MuSEI, transformer, and 1DCNN networks ultimately converge to comparable levels. However, the MuSEI model achieves the highest recognition accuracy, suggesting a relatively lower level of overfitting to the data.

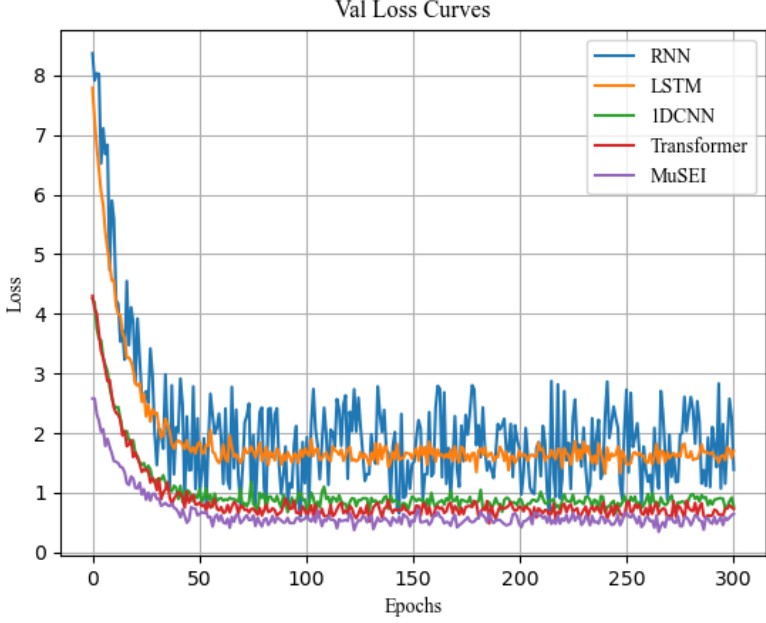

**Figure 9.** The loss curves.

Here, we compare our proposed method with some multimodal networks, such as MuSELI [15] and DCTM [16]. In this experiment, we take the time-domain radar signal

as one modality input and the feature vector as the other modality input—both use data augmentation techniques—and we use cross-entropy as the loss function. The results are shown in Table 4. MuSELI was proposed by Bharadwaj et al., who delved into the use of various metadata sources to enhance language identification. In MuSELI, the metadata text sequence **T** is input to the text encoder, which consists of a token embedding layer to generate the latent representation [15]. However, considering the difference between feature vectors and text signals, we opted not to employ the text encoder in our experiment. Instead, we directly fed the feature vectors and time-domain radar signal representations into the shared encoder. DCTM proposed an architecture for engagement estimation that combines dilated convolution and transformers, which leverages signals from three modalities—speech, pose, and face—as inputs, integrating them via the dilated convolution block. However, in this specific experiment, only the radar's time-domain signal is fed into the dilated convolution block, followed by fusion strategies involving the feature vector. Additionally, the experiment in [16] compares two types of fusion strategies: namely, "self-attention fusion (SA)" and "multimodal gated fusion (GF)". In this experiment, we also compare them. The input for these models is the sequence of radar time-series data; all other methods remain unchanged. MuSEI and MuSELI achieve higher identification accuracy both with and without data augmentation compared with the other models. This shows the effectiveness of the raw data embedding module, which consists of the conformer blocks. Moreover, this reaffirms the efficacy of our proposed data augmentation technique, underscoring its ability to enhance the model's robustness. DCTM might prove to be better suited for representing speech signals, pose signals, and other analogous data. Simultaneously, in the experiments in [16], distinct regressors such as transformer and LSTM were utilized. The divergent performances observed when implementing the GF and SA feature fusion strategies across these models suggest potential variations in their applicability, a discussion we opt not to pursue in this context.

**Table 4.** Comparison of multimodal network performance.

| Method | With Data Augmentation | Without Data Augmentation |
|---|---|---|
| MuSEI | **96.67** | 92.01 |
| MuSELI | 96.10 | 91.87 |
| DCTM + SA | 93.26 | 90.97 |
| DCTM + GF | 90.35 | 87.66 |

To further simulate the influence of different datasets and consider the influence of various additive noises on the signal, we artificially add additive Gaussian white noise to the simulated samples in the range of 5 dB–10 dB. Dataset II contains 10 classes, for which the simulation parameters are different signal types, pulse widths, rising edge slopes, rising edge times, etc. Figure 10 illustrates the identification results as the signal-to-noise ratio changes. The experiment in Figure 10 uses the CSKD loss but not data augmentation. As seen from Figure 10, our method achieve better results compared with MuSELI and DCTM + SA. As the signal-to-noise ratio increases, our method improves the recognition accuracy by nearly 11%. The recognition performance of MuSELI and DCTM + SA for dataset II is relatively similar. Figure 10 demonstrates that although our proposed method might be affected by additive noise, the effect is still less than that of other advanced multimodal methods in the case of a low signal-to-noise ratio.

Table 5 demonstrates the effectiveness of different modules on MuSEI. To assess the efficacy of each module, we conducted ablation experiments on the model. The initial scenario involved substituting MuSEI's embedding module with the conventional transformer embedding module; that is, the traditional embedding module in the transformer replaces the raw data embedding module in Figure 1, denoted as "*traditional embedding". Subsequently, the second scenario utilized the final layer output of the conformer directly as the feature representation; that is, the weighted representation substitutes for the last layer representation, labeled as "*final Conformer" Single-modal input means we only feed

raw time-domain signals into the network. All experiments in Table 5 use the CSKD loss function and data augmentation techniques.

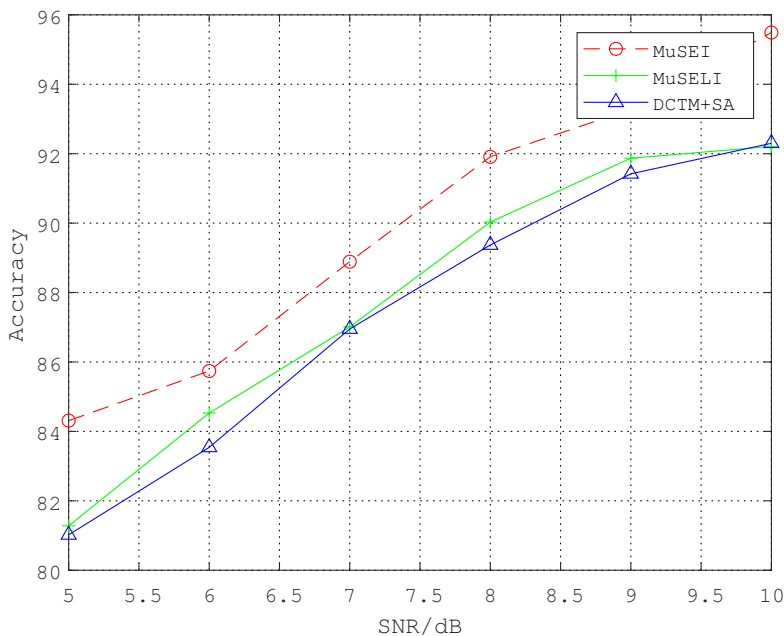

**Figure 10.** Recognition accuracy under different signal-to-noise ratios.

**Table 5.** Ablation experiments on MuSEI.

| Method | Multimodal Input | Single-modal Input |
|---|---|---|
| *traditional embedding | 93.81 | 91.91 |
| *final Conformer | 94.40 | 93.87 |
| Ours | 96.67 | 94.53 |

In the single-modal model, these methods signify the processing of the original signal embedding without generating any concatenated representations. The results in Table 5 suggest the advantage of our data embedding module and multimodal representation module. The conformer module implements global and local interactions through a combination of self-attention and convolution. Moreover, the weighted representation scheme performs more efficiently than using the final layer directly. The network employing the traditional transformer embedding module attained an accuracy of 93.81%, exhibiting about a 3% decrease compared to ours. This illustrates the limitation of traditional embedding in striking a balance between capturing global and local features.

### 4.2.3. Experiment 3: Loss Function Comparison

Table 6 presents the results of our comprehensive experiments conducted with different loss functions. All models in Table 6 use data augmentation techniques. The cross-entropy loss quantifies the divergence between two probability distributions and is typically used in classification tasks. The softmax function transforms raw output scores into probabilities for multiple classes. The maximum entropy principle aims to find the most uncertain probability distribution given a set of constraints or information. Table 6 shows that CSKD outperforms other loss functions on the real radar dataset. We also observe that the top-1 accuracies of other loss functions are often worse than the cross-entropy loss. Compared to the top-1 accuracy of 95.71% achieved by cross-entropy, CSKD elevated it to 96.67%.

**Table 6.** Loss comparison.

| Model | Loss Function | Top 1 (%) |
|---|---|---|
| MuSEI | Cross-entropy | 95.71 |
| | Softmax | 94.89 |
| | Maximum-entropy | 93.76 |
| | CSKD | 96.67 |

Figure 11 visualizes the feature spaces of raw data and MuSEI through t-SNE analysis [39], where 0–5 represent 6 classes of emitters. T-SNE is a visualization technique that can map high-dimensional data into a lower dimensional space while preserving the distances between similar sample points in both spaces. Figure 11 visually demonstrates MuSEI's efficacy in SEI tasks. Compared with the original data distribution, MuSEI can achieve the goal of reducing intra-class variance and increasing inter-class distance to generate more-distinct feature distributions and well-defined boundaries.

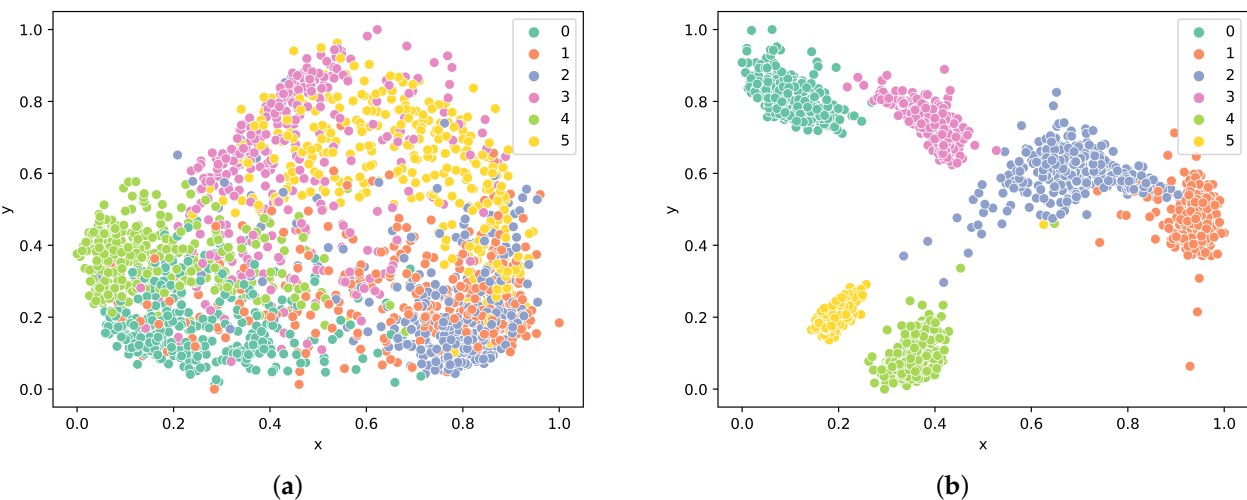

(**a**)          (**b**)

**Figure 11.** Feature visualization. (**a**) The raw data distribution. (**b**) The feature distribution of MuSEI.

## 5. Conclusions

In this paper, we propose the multimodal model MuSEI based on an improved transformer for SEI tasks. MuSEI utilizes raw radar data as well as feature data as multi-modal input. We highlight the benefits of utilizing multimodal data for SEI. The conformer block can learn both position-wise local features and use content-based global interactions. The multimodal representation module has been proven effective at mixing and finding the most informative components. CSKD can effectively mitigate overfitting. Moreover, "CV-CutMixOut" has demonstrated its effectiveness at enhancing data robustness. Extensive experiments on a real radar dataset show that MuSEI outperforms several existing methods. In the future, we will explore the processing of multipath problems[40,41] in real radar data to obtain more effective input data. Meanwhile, we will further investigate the application of multimodal models in SEI, such as combining semi-supervised methods[42].

**Author Contributions:** Conceptualization, H.P. and K.X.; methodology, H.P.; software, H.P.; validation, W.Z.; formal analysis, H.P. and K.X.; investigation, H.P.; resources, K.X.; data curation, W.Z.; writing—original draft preparation, H.P.; writing—review and editing, H.P., K.X. and W.Z.; visualization, H.P.; supervision, K.X. and W.Z.; project administration, K.X.; funding acquisition, K.X. All authors have read and agreed to the published version of the manuscript.

**Funding:** This research was funded by the Guangdong Basic and Applied Basic Research Foundation under grants No. 2021A1515010768 and No. 2023A1515011588 and the Shenzhen Science and Technology Program under grants No. 202206193000001 and No. 20220815171723002.

**Data Availability Statement:** The data utilized in this experiment have been exclusively collected by the research team for internal research purposes. They are not currently available for external sharing or distribution.

**Acknowledgments:** The authors would like to thank the anonymous reviewers, the Associate Editor, and the Editor for their constructive comments and suggestions, which have greatly improved this paper.

**Conflicts of Interest:** The authors declare no conflicts of interest.

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
