# Peer review of "Research on an Enhanced Multimodal Network for Specific Emitter Identification"

_electronics, doi:10.3390/electronics13030651_

Round 1
Reviewer 1 Report
Comments and Suggestions for Authors
In general, the topic of SEI is important and necessary, however, the paper may need more clarification and verifications.
First, why the multimodal is a good/necessary method to be considered? Since we already have feature fusion, and deep learning methods (autoencoder, etc.), thus the reason that "existing multimodal methods are still rarely used in SEI" is not a valid argument to prove the multimodal is necessary and important for SEI.
Second, some papers may have considered using multimodal for SEI, e.g., [3] in the reference list. It would that be better to compare the advanced ML methods not only the classical deep learning methods such as LSTM, RNN. Another example is Dr. Tim Oshea's publications, such as [r1] Semi-Supervised Radio Signal Identification, etc., [r2] radio signal identification, identification system learning, and identifier deployment.
Some minors:
extracting characteristics -> extracting features,
what do you mean: "Prior techniques have been constrained to a single modality," is this a true/solid claim? Some papers may considered as well.
3. Method. The one is to introduce -> the first one is to...
The following paper may be helpful:
[r3] Zhu, Mingzhe, et al. "Compressed sensing mask feature in time-frequency domain for civil flight radar emitter recognition." 2018 IEEE International Conference on Acoustics, Speech and Signal Processing (ICASSP). IEEE, 2018.
Comments on the Quality of English LanguagePlease find the above comments.
Reviewer 2 Report
Comments and Suggestions for Authors
The authors present a method that utilizes the multimodal sources of information to solve the emitter identification problem.
1 The authors show the benefit of multimodal embedding in section 4.2.2 and show that by using the raw signal and utilize the statistical features. The gain in using the multimodal embedding compared to utilizing the raw-signal alone seems to be minimal as shown in Table 5. Can the authors compare the performance with the case when statistical features alone are used for the identification purpose?
2. The authors also have not provided the details of the dataset under consideration. Can the authors shed some light on the type of radar signals considered to help the reader understand?
3. The authors have not given the details if they have considered additive noise or channel effects such as multi-path and fading during the training procedure. Can the authors comment on how the proposed method can be applied to realistic measurements ?
Round 2
Reviewer 1 Report
Comments and Suggestions for Authors
I have no further comments, the authors have addressed the concerns.